

**Heterogeneous reaction of HO₂ with airborne TiO₂ particles and its implication for**
**climate change mitigation strategies**
Daniel R. Moon[1], Giorgio S. Taverna[2], Clara Anduix-Canto[1], Trevor Ingham[1,3], Martyn P. Chipperfield[2,3],
Paul W. Seakins[1,3], Maria-Teresa Baeza-Romero[4], Dwayne E. Heard[1,3] [*]
[1] *School of Chemistry, University of Leeds, Leeds, LS2 9JT, UK*
[2] *School of Earth and Environment, University of Leeds, LS2 9JT, UK*
[3] *National Centre for Atmospheric Science, School of Chemistry, University of Leeds, Leeds, LS2 9JT, UK*
[4] *Escuela de Ingeniería Industrial, Universidad de Castilla-La Mancha, 45071, Toledo, Spain*
[*] Corresponding author. Email: d.e.heard@leeds.ac.uk
**Abstract.**
One geoengineering mitigation strategy for global temperature rises resulting from the increased concentrations of
greenhouse gases is to inject particles into the stratosphere to scatter solar radiation back to space, with $TiO_2$ particles
emerging as a possible candidate. Uptake coefficients of $HO_2$, $\gamma(HO_2)$, onto sub-micrometre $TiO_2$ particles were
measured at room temperature and different relative humidities (RH) using an atmospheric pressure aerosol flow
tube coupled to a sensitive $HO_2$ detector. Values of $\gamma(HO_2)$ increased from $0.021 \pm 0.001$ to $0.036 \pm 0.007$ as the RH
was increased from 11% to 66%, and the increase in $\gamma(HO_2)$ correlated with the number of monolayers of water
surrounding the $TiO_2$ particles. The impact of the uptake of $HO_2$ onto $TiO_2$ particles on stratospheric concentrations
of $HO_2$ and $O_3$ was simulated using the TOMCAT three-dimensional chemical transport model. The model showed
that by injecting the amount of $TiO_2$ required to achieve the same cooling effect as the Mt. Pinatubo eruption,
heterogeneous reactions between $HO_2$ and $TiO_2$ would have a negligible effect on stratospheric concentrations of
$HO_2$ and $O_3$.



## 1. Introduction

It has been suggested that injection of aerosols into the stratosphere in order to scatter solar radiation back to space could be a possible solar radiation management scheme (Shepherd and Working Group on Geoengineering the Climate, 2009). Such a scheme would have the effect of cooling the Earth's surface and serve as a measure to mitigate enhanced global warming. A possible candidate for stratospheric injection is sulphuric acid aerosols as these aerosols occur naturally in the stratosphere. These aerosols are formed by transport of precursors, e.g. $SO_2$ and carbonyl sulphide (OCS), from the troposphere to the stratosphere (Holloway and Wayne, 2010), which form sulphuric acid vapour that condenses onto particles. However, injection of these particles could have important negative effects on the stratosphere through enhanced ozone depletion. Concentrations of sulphate aerosols can increase dramatically as the result of volcanic eruptions. Following the eruption of Mt. Pinatubo in 1991, it was estimated that around 30 Tg of $H_2SO_4$ was injected in the stratosphere, dramatically increasing stratospheric aerosol loading and hence the available surface area for heterogeneous chemistry to occur. Following this volcanic event, the average global lower tropospheric temperature decreased by 0.5 K (Dutton and Christy, 1992), however, stratospheric ozone concentrations reached a record low in northern mid-latitudes (Dutton and Christy, 1992;McCormick et al., 1995) showing sulphate aerosols to be unsuitable for solar radiation management. Other candidates for particle types, such as $TiO_2$, have been put forward due to their large refractive indices (Pope et al., 2012), meaning that less stratospheric aerosol loading would be necessary to achieve the same level of cooling. The refractive index of $TiO_2$ at 550 nm is 2.5 compared to a value of 1.5 for naturally occurring sulphate aerosols (Tang et al., 2014). Assuming that the size of $TiO_2$ particles can be optimised, it has been reported that to achieve the same cooling effect that sulphate aerosols had during the Mt. Pinatubo event, approximately three times less in mass, and seven times less in volume of $TiO_2$ would be required compared with sulphuric acid (Pope et al., 2012). However, the impacts of the presence of $TiO_2$ particles on stratospheric chemistry have to be determined before this kind of geoengineering solution can be considered. Mineral dust particles are commonly found in the troposphere and contribute the largest fraction to tropospheric aerosol loading in terms of mass (Textor et al., 2006;Huneeus et al., 2011). Typically $TiO_2$ (which is classified as a mineral) constitutes from 0.1% to 10% of overall atmospheric mineral dust loading depending on the location of sources (Usher et al., 2003;Karagulian et al., 2006).

The heterogeneous chemistry of sulphate aerosols in the stratosphere is fairly well understood (Ammann et al., 2013); for example the conversion of $NO_x$ to nitric acid in the aerosol via $N_2O_5$ adsorption and reaction, and also the activation of chlorine via the reaction of $ClONO_2$ with HCl to form $Cl_2$ and nitric acid within cold aerosols. However, the heterogeneous reactivity of mineral particles, in particular $TiO_2$, is not as well understood. Removal and production of trace gases in the stratosphere may significantly perturb concentrations of $O_3$, therefore it is important when assessing the potential impact of such a solar radiation management scheme to evaluate the kinetics of likely heterogeneous chemistry.

The hydroperoxyl radical, $HO_2$, is an important species within the stratosphere, being present at about 5 parts per trillion per unit volume (pptv) around the tropopause, and is involved in a $HO_x$ catalytic cycle responsible for about 40% of $O_3$ depletion in the lower stratosphere via to the following reactions (Wennberg, 1994):



$$OH + O_3 \rightarrow HO_2 + O_2 \hspace{4cm} (R1)$$
$$HO_2 + O_3 \rightarrow OH + 2O_2 \hspace{4cm} (R2)$$
$$Net: 2O_3 \rightarrow 3O_2$$
Moreover, $HO_2$ can also react with stratospheric ClO and BrO to produce HOCl and HOBr respectively, which can
be photolysed to produce further OH and atomic halogen species that can contribute to $O_3$ loss. $HO_2$ can undergo
self-reaction upon surfaces of mineral dust, which is thought to result in the generation of $H_2O_2$ (Matthews et al.,
2014), whose two predominant removal pathways are photolysis and reaction with OH (Versick et al., 2012):
$$H_2O_2 + h\nu \ (\lambda \leq 360 \ nm) \rightarrow 2OH \hspace{3cm} (R3)$$
$$H_2O_2 + OH \rightarrow HO_2 + H_2O \hspace{3.5cm} (R4)$$
Although the kinetics of the uptake of $HO_2$ onto Arizona Test Dust (ATD), a proxy of mineral dust, has been
previously investigated (Matthews et al., 2014;Bedjanian et al., 2013) the heterogeneous reaction of $HO_2$ with $TiO_2$
has not been studied. However, the kinetics of $N_2O_5$ uptake (Tang et al., 2014) and $ClONO_2$ (Tang et al., 2016) onto
$TiO_2$ have been studied. The heterogeneous reaction of $N_2O_5$ results in the conversion to reactive nitrogen oxides
(NO and $NO_2$) involved in a catalytic cycle that leads to significant $O_3$ depletion and non-reactive $HNO_3$. The reactive
uptake coefficient, $\gamma$, which is the probability that a species will collide with an aerosol and be removed by reaction,
was measured to be more than an order of magnitude larger for $HO_2$ onto ATD than for $N_2O_5$ and $ClONO_2$ onto $TiO_2$
sub-micron particles, and contrasting dependences of $\gamma$ with relative humidities (RH) were observed. Therefore, by
analogy it is expected that $HO_2$ uptake onto $TiO_2$ may be faster than $N_2O_5$ uptake. $ClONO_2$ uptake by $TiO_2$ particles
resulted in similar values of $\gamma$, however, no dependency of $\gamma$ with RH between 7 – 33% was observed.
In this investigation an aerosol flow tube coupled with a sensitive $HO_2$ detector based on chemical conversion
followed by laser-induced fluorescence detection of OH (George et al., 2013) has been used to study the kinetics of
the heterogeneous reaction of $HO_2$ with airborne $TiO_2$ nanoparticles at different RH. While it has been possible here
to study such kinetics over a range of RH representative to those typically found in the lower stratosphere (< 40%)
(Wennberg et al., 1994), experimental limitations meant that experiments were only conducted at room temperature
(~293 K). The TOMCAT off-line three-dimensional (3D) chemical transport model (Chipperfield; 1999) has also
been used to predict the likely impact of $HO_2$ uptake by $TiO_2$ particles on the stratospheric concentrations of $HO_2$
and $O_3$.

## 91 2. Methods.

### 92 2.1. Overview of experimental apparatus

The experimental setup deployed for this investigation is similar to other investigations of $HO_2$ uptake by aerosols
undertaken at the University of Leeds (George et al., 2013;Matthews et al., 2014;Lakey et al., 2016) therefore a





detailed description of the components of the experiment is not given.  A schematic diagram of the experiment is
shown in Figure 1, and all experiments were undertaken at room temperature (~293 K) and under normal laboratory
levels of illumination. For some experiments the flow tube was covered with a black shield to eliminate light and no
differences in the results were observed.

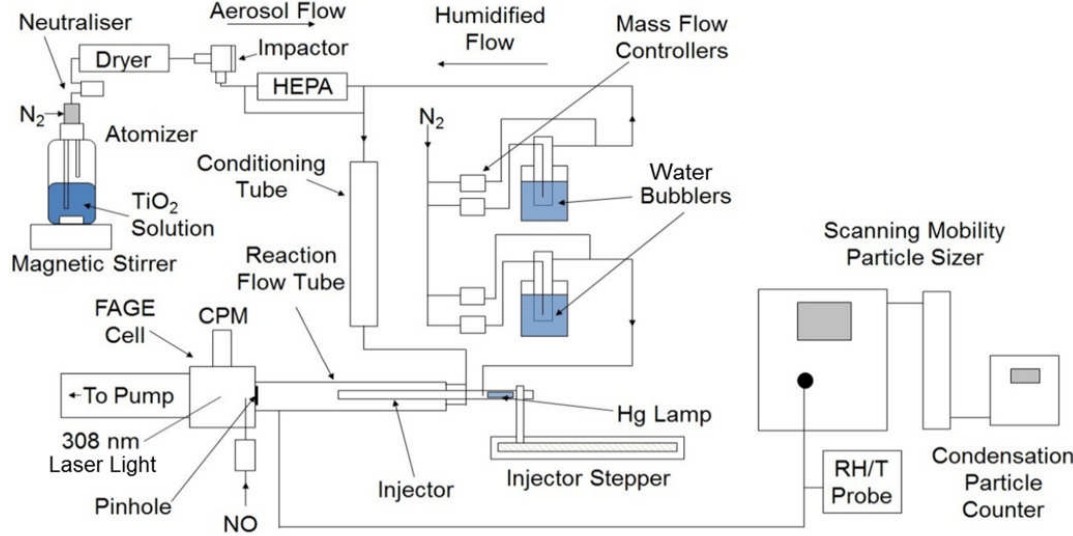


**Figure 1**. Schematic diagram of the aerosol flow tube experiment. CPM: Channel Photomultiplier, HEPA: high-
efficiency particulate air filter, FAGE: fluorescence assay by gas expansion, RH/T: Relative Humidity/Temperature.

Compressed nitrogen, which had been passed through a gas purification system (TSI 3074B) consisting of particle
filters, a dryer and a carbon filter, was used as the carrier gas for the experiments.  A humidified flow of $TiO_2$ particles
was introduced through two inlets located at the rear of the aerosol flow tube and the flow of $HO_2$ radicals enters the
flow tube via the movable injector.  The total flow through the flow tube (107 cm length, 5.9 cm I.D.) was 5.4 L
$min^{-1}$ which resulted in a Reynolds number of 130, and therefore is considered laminar as confirmed by radial
concentration gradient measurements of gases exiting the injector (George et al., 2013).  Experiments consisted of
moving the injector using a linear drive (BSL Engineering 15 KR4610A) to different fixed positions along the flow
tube (30-70 cm from injector tip to $HO_2$ detector inlet in steps of 5 cm) corresponding to reaction times between ~ 8
and 20 s from the injector, with detection of $HO_2$ at the end of the flow tube.  All gas flows within the experiment
were controlled using mass flow controllers (Brookes and MKS).  The RH of the flow was measured using a probe
(Rotronics) in the exhaust of the flow tube, which itself was calibrated against a dew point hygrometer (Buck
Research Instruments CR-4).





## 2.2. Aerosol Generation and Detection

A solution of $TiO_2$ (Aldrich Chemistry 718467, 99.5% Degussa, 5 g in 500 ml of Milli-Q water) was placed in a commercial atomizer (TSI 3076) in order to produce a 1 L min$^{-1}$ flow entrained with $TiO_2$ particles, referred to as the aerosol flow. The aerosol flow was then passed through a neutraliser (Grimm 5522) to reduce static wall losses, a diffusion drier (TSI 3062) and an impactor (TSI 1035900) to ensure larger aerosols, beyond the detection range of the SMPS (~ 750 nm diameter), do not enter the aerosol flow tube. A high-efficiency particulate air (HEPA, PALL Life Sciences) filter situated within a by-pass loop was used to control the number concentration of particles entering the aerosol flow tube. The aerosol flow was then mixed with a humidified flow of nitrogen (3 L min$^{-1}$) to control the RH within the system. The RH of the humidified flow was altered by changing the ratio of dry nitrogen and nitrogen passed through a water bubbler. This flow was then passed through a conditioning tube (residence time ~ 6 s) before entering the aerosol flow tube to allow time for water adsorption onto the surface of the $TiO_2$ particles to equilibrate at the given RH.

The total surface area of $TiO_2$ particles available for heterogeneous reaction with $HO_2$ was measured with a SMPS instrument from the flow exiting the aerosol flow tube. Previous experiments showed that there is a negligible loss of aerosols during the transit of the flow tube (George et al., 2013). The SMPS consisted of a Differential Mobility Analyzer (DMA, TSI 3080, 3081) that creates a monodisperse flow of aerosols based on their electrical mobility which is related to their size. A condensation particle counter (CPC, TSI 3775) connected in parallel to the DMA quantified particle number concentrations. These two instruments connected in parallel can be used to create an aerosol size distribution from which the total surface area and average radius of particles can be calculated by making the assumption that particles are spherical (as demonstrated experimentally in section 3.2 below). The average diameter of the particles is 136 nm and 173 nm at RH = 11% and 37%, respectively.

## 2.3 $HO_2$ Generation and Detection

$HO_2$ radicals were produced within the movable injector (110 cm length, 1.9 cm O.D., 1.6 cm I.D.) by passing a 1.3 L min$^{-1}$ humidified flow of nitrogen (consisting of a mixture of 0.9 L min$^{-1}$ of dry $N_2$ and 0.4 L min$^{-1}$ $N_2$ passed through a water bubbler) containing trace amounts of oxygen over a mercury lamp (L.O.T.-Oriel 6035) via the following reactions:

$$H_2O + h\nu \ (\lambda = 185 \ nm) \rightarrow OH + H \quad\quad\quad (R5)$$

$$H + O_2 + M \rightarrow HO_2 + M \quad\quad\quad (R6)$$

OH is also created by the photolysis of water vapour in R5, but no OH was observed exiting the injector, presumably owing to rapid losses at the walls of the injector. $HO_2$ was sampled by a 0.7 mm diameter pinhole at the end of the flow tube, and after chemical conversion to OH by addition of excess NO (50 sccm, BOC, 99.5 %) just inside the pinhole, laser induced fluorescence (LIF) at low-pressure (the fluorescence assay by gas expansion (FAGE) technique (Heard and Pilling, 2003)) was used to measure OH. The relative LIF signal from converted $HO_2$ was calibrated using an established method (Winiberg et al., 2015) developed for field measurements of OH and $HO_2$ radicals. Hence the experiment is able to measure the absolute concentration of $HO_2$ during passage from the injector to the





sampling inlet. The $Q_1(2)$ line of the OH ($A^2\Sigma^+$ $v' = 0 - X^2\Pi_i - v'' = 0$) transition at ~ 308 nm was used to detect OH.
A Nd:YAG pumped dye laser (JDSU Q201-HD Q-series, Sirah Cobra Stretch) was used to produce the required 308
nm radiation (line width ~ 0.1 cm$^{-1}$) at a pulse repetition rate of 5 kHz. As the flows through the movable injector
(1.3 L min$^{-1}$) and mercury lamp current (20 mA) were kept constant, it is assumed that the initial $HO_2$ concentration,
$[HO_2]_0$ (defined in this investigation as $[HO_2]$ at the first injector position, i.e. 30 cm downstream of the injector),
determined by calibration to be $1.6 \times 10^9$ molecule cm$^{-3}$, was the same for all experiments. These $HO_2$ concentrations
are ~ 50 times higher than typical levels in the sunlit stratosphere (Wennberg et al., 1994). A reference fluorescence
cell, in which a large concentration of OH was generated and detected by LIF, was used to facilitate the identification
of OH lines and tune the laser wavelength. The FAGE cell was continuously evacuated using a combination of a
rotary pump (Edwards, model E1M80) and a roots blower (EH1200), and was kept at 0.8–0.9 Torr, which was
monitored using a capacitance monitor (Tylan General, CDC 11).
**2.4. Experimental Procedure and Data Analysis**
The $HO_2$ signal was measured at 8 positions as the moveable injector was drawn back from 30 to 70 cm using the
linear stepper drive and again as the moveable injector was pushed forwards back to its initial position. The $HO_2$
signal was averaged over 20 s (average of twenty 1 s measurement points, each corresponding to 5000 laser shots) at
each injector position with a 22 s delay between measurements at each injector position in order to allow time for
mechanical vibrations to subside, and to ensure a full flush of the aerosol flow tube so that the LIF signal corresponds
to $HO_2$ emitted from the injector position being measured. The laser power was recorded for each injector position
and used to normalise the $HO_2$ signal to correct for any fluctuations in laser power (< 5%). The $HO_2$ signals with the
injector moving forwards and backwards were then averaged, and this procedure repeated six times with varying
concentrations of aerosols present in the aerosol flow tube. The wall loss rate of $HO_2$ ($k_{wall}$) was determined by
recording the $HO_2$ decay in the absence of aerosols, but at the same RH, and was repeated four times for each
experiment.
The $HO_2$ concentration as a function of time along the flow tube can be expressed as:
$$\ln[HO_2]_t = \ln[HO_2]_0 - k_{obs}\, t \qquad \text{(Eqn. 1)}$$

where $[HO_2]_t$ and $[HO_2]_0$ are concentrations of $HO_2$ at time $t$ and $t = 0$ respectively, and $k_{obs}$ is the observed pseudo-
first-order rate coefficient for $HO_2$ uptake. As the $HO_2$ signal is directly proportional to the concentration of $HO_2$,
the gradient of a plot of $\ln(HO_2$ Signal) against time (calculated from the injector position and measured flow rate)
yields $k_{obs}$, as shown in Figure 2.



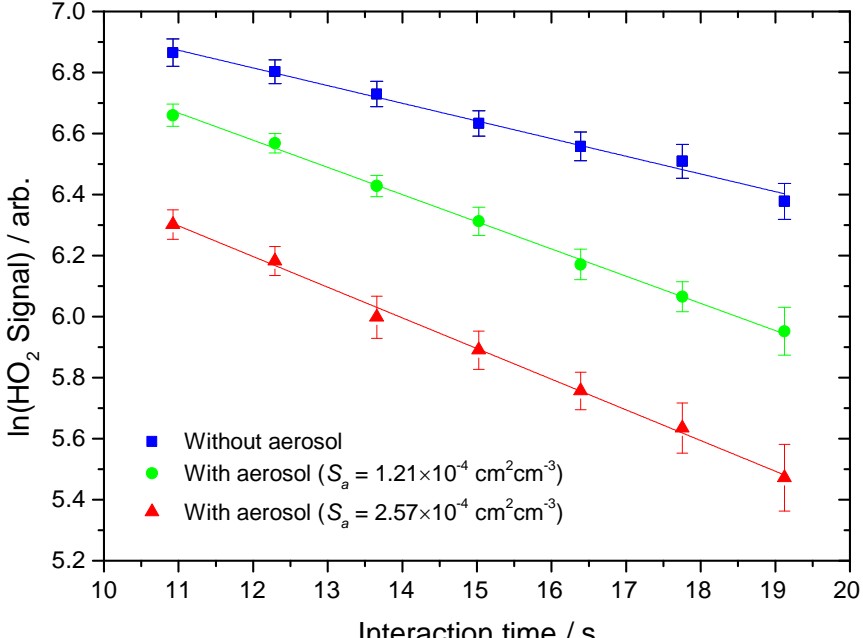


**Figure 2.** Measured $HO_2$ signal at different interaction times, in the presence of $TiO_2$ particles (surface area density $1.21 \times 10^{-4}$ cm$^2$ cm$^{-3}$ and $2.57 \times 10^{-4}$ cm$^2$ cm$^{-3}$, green circles and red triangles respectively) and in their absence (blue squares). The RH in the aerosol flow tube was 11%, the lowest used in this study. The lines represent linear-least squares fits to the data yielding $k_{obs} = 0.079 \pm 0.005$ s$^{-1}$ and $k_{obs} = 0.093 \pm 0.003$ s$^{-1}$ (aerosols present, green circles and red triangles respectively) and $0.049 \pm 0.003$ s$^{-1}$ (no aerosols, $k_{wall}$).

The flow tube was coated with halocarbon wax (Halocarbon Products Corporation, Halocarbon Wax Series 600) to reduce $HO_2$ wall loss rate ($k_{wall}$) and an average of $k_{wall}$ from several determinations in the absence of aerosols was subtracted from $k_{obs}$. A correction (typically about 30%) to account for non-plug flow conditions in the aerosol flow tube using the procedure outlined by Brown (Brown, 1978) was applied to yield, $k'$, the pseudo-first-order loss of $HO_2$ by heterogeneous reaction with $TiO_2$ particles. The relationship between $k'$ and total surface area of $TiO_2$ particles ($S_a$) can be expressed as (George et al., 2013):

$$k' = \frac{w_{HO_2} \gamma_{obs} S_a}{4} \qquad \text{(Eqn. 2)}$$

where $w_{HO2}$ is the mean velocity of $HO_2$ (~ 435 ms$^{-1}$ at 293 K) and $\gamma_{obs}$ is the observed reactive uptake coefficient, obtained from a plot of $k'$ versus $S_a$, an example of which is given in Figure 3. $\gamma_{obs}$ was in turn corrected (typically ~1%) to account for the gas diffusion limitation (Fuchs and Sutugin, 1970), to yield $\gamma(HO_2)$.





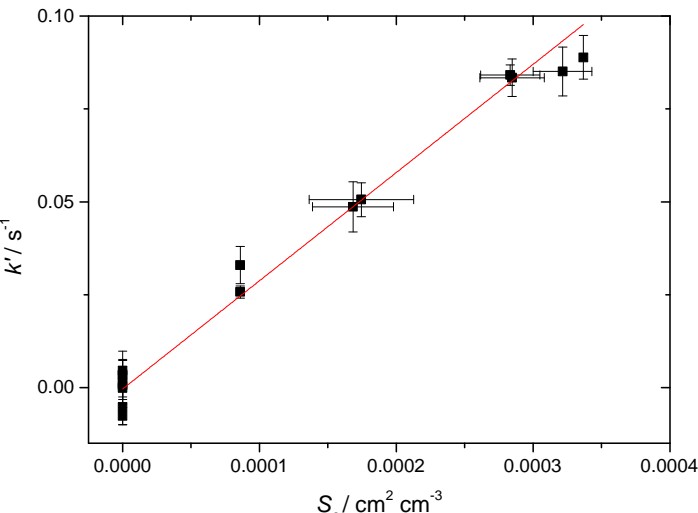

196

**Figure 3**. First order rate coefficient for loss of $HO_2$ due to heterogeneous reaction with $TiO_2$ particles at different

total surface areas for T = 296 K and RH = 37%. The gradient yielded a value of $\gamma(HO_2) = (2.68 \pm 0.01) \times 10^{-2}$, with

the uncertainty representing $2\sigma$ random errors from the fit (95% confidence limits).

## 2.5. TOMCAT Model Description

The TOMCAT off-line three-dimensional (3D) chemical transport model (CTM) (Chipperfield, 1999, 2006) has been
used to predict the impact of the heterogeneous reaction of $TiO_2$ with $HO_2$ to stratospheric concentrations of $O_3$ and
$HO_2$. The model has been widely used in previous studies of stratospheric chemistry and performs well in reproducing
stratospheric ozone and the trace species which control is distribution (Chipperfield et al., 2015). The model includes
a detailed treatment of stratospheric chemistry of $O_x$, $HO_x$, $NO_y$, $Cl_y$ and $Br_y$ species along with the main source
gases. The model has a comprehensive gas-phase chemistry scheme and includes a number of heterogeneous
reactions on stratospheric sulphate aerosols and polar stratospheric clouds (Chipperfield, 1999).
The loss rate of $HO_2$ due to heterogeneous reaction with $TiO_2$ was included in the model as:
$$k = 0.25\, S_a\, w_{HO2}\, \gamma(HO_2) \qquad \text{(Eqn. 3)}$$
where $S_a$ is the surface area density of $TiO_2$, and $w_{HO2}$ and $\gamma(HO_2)$ are defined above. Three TOMCAT simulations
were performed at a horizontal resolution of $5.6^{\circ} \times 5.6^{\circ}$ and 32 levels from the surface to ~ 60 km. The model was
forced with wind and temperature fields from the European Centre for Medium-Range Weather Forecasts (ECMWF)
ERA-Interim reanalyses and integrated for 2 years from January 2007 until December 2008, initialised with the
output from a standard TOMCAT run which had spun-up from 1977. More information on the model experiments is
given in Section 3.3.




## 3. Results and Discussion

### 3.1. The reactive uptake coefficient dependence with relative humidity

The reactive uptake coefficient for $HO_2$ radicals, $\gamma(HO_2)$, onto $TiO_2$ particles was determined at eight different RH (11 - 66%), as shown in Figure 4 and summarised in Table 1. The number of monolayers of water adsorbed onto the surface of $TiO_2$ particles has been previously determined experimentally by transmission FTIR spectroscopy (Goodman et al., 2001), and is also shown as a function of RH in Figure 4.

**Table 1**. Reactive $HO_2$ uptake coefficients, $\gamma(HO_2)$, for $TiO_2$ particles at different RH.

| RH (%) ± 1.0 | $\gamma(HO_2) \times 10^{-2}$ |
|---|---|
| 11.1 | 2.08 ± 0.11 |
| 11.8 | 2.11 ± 0.13 |
| 24.9 | 2.48 ± 0.33 |
| 29.1 | 2.54 ± 0.18 |
| 37.5 | 2.68 ± 0.09 |
| 45.1 | 2.90 ± 0.53 |
| 45.3 | 2.92 ± 0.37 |
| 63.1 | 3.43 ± 0.27 |
| 66.4 | 3.65 ± 0.70 |

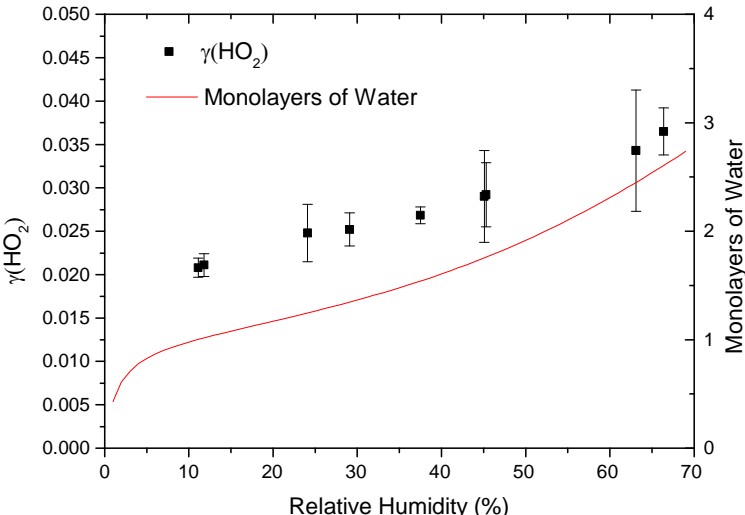

**Figure 4**. Reactive uptake coefficients of $HO_2$, $\gamma(HO_2)$, onto airborne $TiO_2$ particles (black squares, left $y$ axis) at different RH for $T = 295 \pm 2$ K. The number of monolayers of the adsorbed water on $TiO_2$ particles (red curve, right $y$ axis) at 296 K is also plotted as a function of RH, determined using FTIR spectroscopy (Goodman et al., 2001).





The results clearly show a positive dependence of $\gamma(HO_2)$ across the range of RH investigated, and as shown in Figure
5, $\gamma(HO_2)$ correlates well with the number of monolayers of water adsorbed onto the $TiO_2$ particles, $V/V_m$, determined
by Goodman *et al.* (2001), and which was parameterised by Eqn. 4:
$$\frac{V}{V_m} = \left[\frac{c\left(\frac{P}{P_0}\right)}{1-\left(\frac{P}{P_0}\right)}\right]\left[\frac{1-(n+1)\left(\frac{P}{P_0}\right)^n + n\left(\frac{P}{P_0}\right)^{n+1}}{1+(c-1)\left(\frac{P}{P_0}\right)-c\left(\frac{P}{P_0}\right)^{n+1}}\right]$$
(Eqn. 4)

where $V$ is the volume of gas (water vapour) adsorbed at equilibrium pressure $P$, $V_m$ is volume of gas necessary to
cover the surface of the adsorbent $TiO_2$ particles with a complete monolayer, $P$ is the equilibrium pressure of the
adsorbing gas, $P_0$ is the saturation vapour pressure of the adsorbing gas at that temperature, $c$ is a temperature-
dependent constant related to the enthalpies of adsorption of the first and higher layers and $n$ is the asymptotic limit
of monolayers (~8).

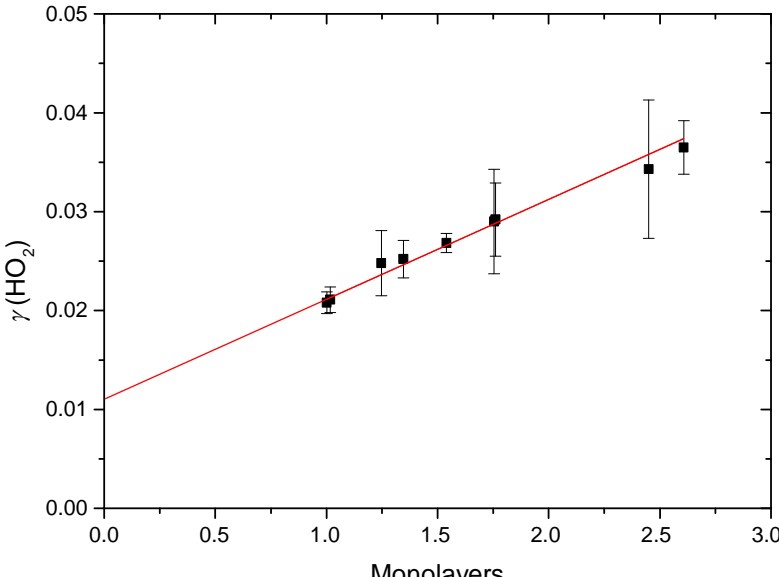


**Figure 5**. Variation of the reactive uptake coefficient, $\gamma(HO_2)$, with the number of monolayers of water surrounding
$TiO_2$ particles (as determined by Goodman et al., 2001) for $T = 295 \pm 2$ K. The red line represents a linear least-
squares fit to the data ($r^2 = 0.987$).

Wall losses of $HO_2$ also increase as RH is increased within the aerosol flow tube, and so in the absence of aerosols,
the $[HO_2]$ for a given distance from the injector will decrease with RH. Previous work in this laboratory has shown
that $\gamma(HO_2)$ for uptake on ATD aerosols increases as the $[HO_2]$ is lowered (Matthews et al., 2014), and hence some
of the positive dependence of $\gamma(HO_2)$ with RH shown in Figure 4 might be expected simply owing to the $[HO_2]$
impinging on the aerosol for a given injector position decreasing with RH. To investigate this further, uptake onto



TiO$_2$ at RH = 11% was measured as a function of [HO$_2$]$_o$ exiting the injector, and $\gamma$(HO$_2$) increased from $2.08 \times 10^{-2}$
to $2.72 \times 10^{-2}$ as [HO$_2$]$_0$ was decreased from $1.6 \times 10^{9}$ molecule cm$^{-3}$ to $8.9 \times 10^{8}$ molecule cm$^{-3}$. However, the wall
loss rate for HO$_2$ only increased from 0.049 to 0.079 s$^{-1}$ across the entire range of RH (11% to 66%) resulting in only
a small change in [HO$_2$], decreases of ~ $2.6 \times 10^{8}$ molecule cm$^{-3}$ and ~ $2.7 \times 10^{8}$ molecule cm$^{-3}$ at the first and last
injector position, respectively. In previous studies of HO$_2$ uptake onto ATD (Matthews et al., 2014) it was shown
that the increase of $\gamma$(HO$_2$) with decreasing [HO$_2$] is linear. Therefore, assuming the same behaviour for uptake onto
TiO$_2$ particles, the expected change in $\gamma$(HO$_2$) as a result of RH increasing from 11-66% due only to a change in
initial HO$_2$ concentration is only $\Delta\gamma$(HO$_2$)=0.0023 (or ~6% of the averaged measured $\gamma$(HO$_2$) across this range).
Hence the 175 % increase in $\gamma$(HO$_2$) observed in Figure 4 across this range of RH is due to another reason.
Figure 5 shows that, for the range of RH studied, $\gamma$(HO$_2$) is a linear function of the number of monolayers of water
surrounding a TiO$_2$ particle, suggesting that water plays a role in the reactive HO$_2$ uptake process on TiO$_2$ particles.
Studies using ambient pressure photoelectron spectroscopy (APPES) (Yamamoto et al., 2008; Ketteler et al., 2007)
have shown that water adsorption on rutile, a polymorph of TiO$_2$ (110), occurs in distinct steps, as illustrated in Figure

262   6.


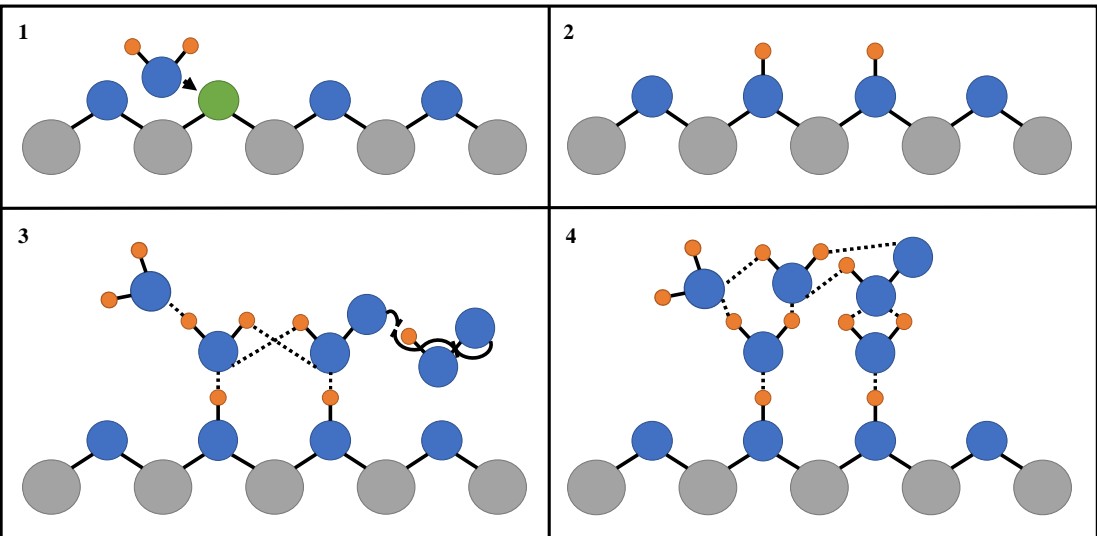

**Figure 6.** Simplified diagram of the important steps of HO$_2$ reactive uptake onto the surface of TiO$_2$. Grey circles
= Ti, blue circles = O, orange circles = H, green circles = V$_{bridge}$, solid black lines = chemical bond and dashed lines
= hydrogen bond. Panel 1 shows the diffusion of water molecule towards a bridging O vacancy. Panel 2 shows the
resultant formation of two neighbouring bridging OH groups at the original site of the O vacancy. Panel 3 shows
the bridging OH groups acting as anchoring sites for water and HO$_2$ adsorption via hydrogen bonding leading to
multilayer water adsorption and HO$_2$ self-reaction via an Eley-Rideal type mechanism. Panel 4 shows the build up
of a more extensive hydrogen bonded network as more water molecules adsorb onto the particle, which stabilises
HO$_2$ increasing its desorption lifetime and hence probability that it will react.





First, water molecules dissociate at O-vacancies ($V_{bridge}$) in bridge sites, producing a stoichiometric amount of
adjacent bridging OH groups ($OH_{bridge}$) equal to twice the initial vacancy concentration upon initial exposure of rutile
to water vapour:
$$H_2O + V_{bridge} + O_{bridge} = 2OH_{bridge} \quad\quad (R7)$$
This step takes place even at very low RH, with these bridging OH groups acting as nucleation sites for subsequent
water adsorption by anchoring water molecules to form strongly bound OH–$H_2O$ complexes. The OH–$H_2O$
complexes continue to act as nucleation centres for further water adsorption. The wetting properties of $TiO_2$ (110)
are thus driven by moderate amounts of strongly attractive OH sites that nucleate water molecules. These OH-$H_2O$
complexes have a relatively high enthalpy of adsorption for water of -72 kJ mol$^{-1}$ (Ketteler et al., 2007), whereas
additional adsorption of water beyond a monolayer of water coverage is more characterised by enthalpies associated
with the bulk enthalpy of water condensation (-45 kJ mol$^{-1}$) (Chen et al., 2012), explaining the variation of the number
of monolayers of water with RH shown in Figure 4. Computational studies (Aloisio and Francisco, 1998) showed
that in the gas-phase $HO_2$ can also readily form complexes with water through hydrogen bonding with a binding
energy of 28.9 kJ mol$^{-1}$. The observed correlation of $\gamma(HO_2)$ with the number of monolayers of water surrounding
$TiO_2$ particles could be explained by two effects. An increase in the network of hydrogen bonding would increase
the stability of a molecular system, ~ 20.9 kJ mol$^{-1}$ for each hydrogen bond (Joshi and Ghanty, 2013), or that simply
more $HO_2$ can adsorb onto the particle surface as the number of available sites for hydrogen bonding increases. An
increase in the adsorption lifetime of $HO_2$ owing to the more extended H-bonding network (Joshi and Ghanty, 2013)
will result in an increased probability of $HO_2$ reacting with another $HO_2^-$, increasing the value of $\gamma(HO_2)$:
$$HO_2 + HO_2 \rightarrow H_2O_2 + O_2 \quad\quad (R8)$$
For up to ~2 monolayers of $H_2O$, an Eley-Rideal (ER) mechanism of a gas phase $HO_2$ molecule reacting an adsorbed
$HO_2$ is more likely than for two adsorbed $HO_2$ molecules diffusing together to react via a Langmuir-Hinshelwood
(LH) mechanism. The coverage of $V_{bridge}$ across the surface of $TiO_2$ (110) is 0.125 monolayers (i.e. one vacancy for
every eight unit cells) (Ketteler et al., 2007) meaning water will initially adsorb onto the surface of the particle in
clusters. As the binding energy of the $HO_2$ water complex is fairly high, the rate of $HO_2$ diffusion across the surface
after the initial adsorption at a bridging site will be slow, making a LH type mechanism seem unlikely. Above ~2
monolayers, the adsorbed water clusters will begin to interact with each other via an extended H-bonded network
more characteristic of bulk liquid water, allowing $HO_2$ to diffuse around the thin surface film of water with
thermodynamic properties similar to liquid water (Ketteler et al., 2007).
The superoxide ion, $O_2^-$, which is the conjugate base of $HO_2$, reacts with $HO_2$ over a hundred times quicker than with
$HO_2$ via:
$$HO_2 + O_2^- + H_2O \rightarrow H_2O_2 + O_2 + OH^- \quad\quad (R9)$$
It is unlikely that $O_2^-$ will form readily on the surface of the particle at low coverages of water. $O_2^-$ can form on
surfaces via direct surface–oxygen electron transfer, photo-induced electron transfer, surface intermolecular electron




transfer or decomposition of hydrogen peroxide ($H_2O_2$) (Anpo et al., 1999). The dissociation of $HO_2$ to $O_2^-$ occurs in
bulk liquid ($pK_a = 4.7$, (Thornton and Abbatt, 2005)), however, water surrounding the particles only begins to acquire
liquid like properties once the coverage of water is greater than ~2 monolayers.  If significant formation of $O_2^-$ does
occur above ~2 monolayers of $H_2O$, then owing to its much higher reactivity, some deviation from linearity of $\gamma(HO_2)$
versus monolayers of water (Figure 5) might be expected. However, this was not observed, although the number of
data points is very limited.

### 3.2. Comparison of $\gamma(HO_2)$ with literature values

There are previous studies of $HO_2$ uptake onto ATD (Matthews et al., 2014), a proxy for mineral dust, and both $N_2O_5$
(Tang et al., 2014) and $ClONO_2$ (Tang et al., 2016) uptake onto $TiO_2$ particles.  Values of $\gamma(HO_2)$ measured for ATD
are comparable to those measured onto $TiO_2$ particles at similar concentrations of $HO_2$ in the aerosol flow tube
($\gamma(HO_2)_{ATD}$ ~ 0.008 – 0.030).  For ATD, $\gamma(HO_2)$ was only determined at 4 values of RH, but a general increase with
RH was observed (although there was a dip around RH = 50% before a further increase), consistent with $HO_2$ uptake
being driven by the number of defects in the crystal lattice of mineral dust surfaces which provides bridging OH
groups upon exposure to water and subsequently by the number of monolayers of water adsorbed onto the surface of
such aerosols. The number of monolayers of water on the ATD surface at different RH has been determined
(Gustafsson et al., 2005), and also showed a general increase with RH but with a shoulder around RH = 50% where
the observed $\gamma(HO_2)$ also contained a small dip.
Values of $\gamma(N_2O_5)$ (Tang et al., 2014a; and corrigendum Tang et al. 2014b) and $\gamma(ClONO_2)$ (Tang et al., 2016) onto
$TiO_2$ particles were an order of magnitude lower than $\gamma(HO_2)$ and may be associated with the lower polarity of these
molecules compared to $HO_2$.  It is likely that, as with uptake of $HO_2$, both $N_2O_5$ and $ClONO_2$ react with $TiO_2$ via
complexing with bridging OH groups and adsorbed $H_2O$, therefore less polar molecules will be less bound to the
surface and more likely to desorb back into the gas phase, leading to a smaller $\gamma$.  The dependence of $\gamma(N_2O_5)$ for
$TiO_2$ with RH is also different to that for $HO_2$ observed here, with a small decrease of $\gamma(N_2O_5)$ observed as the RH is
increased from ~ 5 – 23% where a minimum is reached, then beyond 23% $\gamma(N_2O_5)$ increases as RH is increased.
(Tang et al., 2014).  Competition between water and $N_2O_5$ for surface active OH groups was suggested for the initial
observed decrease in $\gamma(N_2O_5)$ with RH, whereas for RH > ~ 23% heterogeneous hydrolysis of $N_2O_5$ to form $HNO_3$
starts to drive reactive uptake resulting in an increase of $\gamma(N_2O_5)$ (Tang et al., 2014).  Measurements of $\gamma(ClONO_2)$
onto $TiO_2$ particles were only made at two values of RH, insufficient to determine any systematic dependence,
although Tang *et al.* (2016) expected $\gamma(ClONO_2)$ to increase as more water adsorbs onto the surface of $TiO_2$ particles.
George et al. (2013) previously measured $\gamma(HO_2)$ onto dry salt aerosols. Values of $\gamma(HO_2)$ for NaCl and $(NH_4)_2SO_4$
at RH values (33 – 54%) below their deliquescence point were below the limit of detection ($\gamma(HO_2)$ <0.004).  Values
of $\gamma(HO_2)$ for $TiO_2$ reported here (Figure 4 and Table 1) are more than an order of magnitude greater than that of
solid salts, even for RH = 11%.  A possible explanation for the difference in $\gamma(HO_2)$ values could be that even though
the sizes of the aerosols determined by the SMPS are similar, dry salt aerosols are more spherical in shape than $TiO_2$
particles which may be more fractal in nature.  As the SMPS indirectly measures the surface area of aerosols by
measuring their mobility through an electric field, an assumption that the aerosol is spherical has to be made. If this
is not the case, this may lead to a significant under-prediction of the surface area of non-spherical aerosols and
therefore an over-prediction of $\gamma(HO_2)$. In order to measure the geometry of the $TiO_2$ particles, a scanning electron
microscope (SEM, FEI Nova NanoSEM 450) operating at 3 kV was used to image the $TiO_2$ nanoparticles used within
these experiments. Samples for the SEM were prepared by dispersing the nanoparticles in ethanol and allowing a
drop of this solution to dry on a silicon wafer. The wafer with nanoparticles was then mounted on SEM stubs using
conductive carbon tapes and coated with 2 nm of Ir, using a Cressington 208HR high resolution sputter coater. Figure
7 shows an example of an SEM image providing evidence that the $TiO_2$ particles are spherical and therefore any error
associated with SMPS measurements of surface area is minimal. A more likely explanation for the higher $\gamma(HO_2)$
for $TiO_2$ particles is that dry salt aerosols do not adsorb a significant amount of water onto their surface until the
deliquescence point reached, whereas at RH = 11% Figure 4 shows that the $TiO_2$ particles already have a monolayer
coverage which can form relatively strongly bound complexes with $HO_2$. Measurements of $\gamma(HO_2)$ onto aqueous salt
aerosols show that $\gamma(HO_2)$ significantly increases above the deliquescence point (George et al., 2013) and is
comparable to $\gamma(HO_2)$ measured for $TiO_2$ ($\gamma(HO_2)$ ~ 0.01).

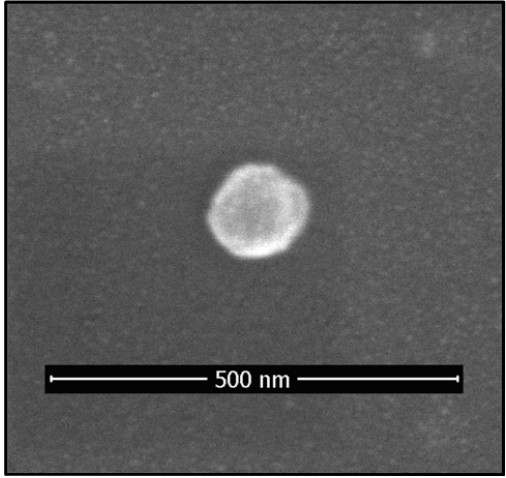

500 nm


**Figure7**. SEM image of a single $TiO_2$ particle magnified 80,000× used within this study.
## 3.3    Implications of $HO_2$ uptake by $TiO_2$ for stratospheric chemistry
The effect of $HO_2$ uptake onto $TiO_2$ particles upon the stratospheric concentrations of $HO_2$ and $O_3$ was assessed using
the TOMCAT model (Chipperfield, 1999, 2006). At RH relevant to the lower stratosphere ($< 40\%$) the measurements
showed that $\gamma(HO_2)$ is in the range 0.020 - 0.028 at 295 K. An inverse temperature dependence of $\gamma(HO_2)$ onto dry
sea-salt aerosols has previously been observed (Remorov et al., 2002), and although there have been no systematic
experimental studies of the temperature dependence of $\gamma(HO_2)$, parameterisations have developed (Thornton et al.,
2008;Macintyre and Evans, 2011). At stratospherically relevant temperatures ($T$ = 200 - 220 K), $\gamma(HO_2)$ is likely to
be considerably larger than observed at 295 K, however it is not possible to cool the aerosol flow tube/SMPS system
to verify this experimentally. Therefore $\gamma(HO_2)$ =1 was used in the model simulations representing an upper limit,





with three TOMCAT simulations performed as follows. A control simulation, similar to that presented in
Chipperfield et al. (2015) did not include $TiO_2$ particles. A specified latitude-height distribution of $TiO_2$ particles was
then included in two simulations with an effective aerosol surface area density equal to that of sulphate aerosols in
1992 the year after the eruption of Mt. Pinatubo. This is an assumption which allows for the fact that less $TiO_2$ mass
is needed in order to produce the same radiative impact as sulphate aerosol from Mt. Pinatubo, but the $TiO_2$ particle
size is smaller. Hence these effects largely cancel (Tang et al., 2014). Stratospheric injection via a geo-engineering
solution will result initially in a different distribution of $TiO_2$ particles compared with after the Mt. Pinatubo eruption,
but it is assumed that following mixing and transport the distributions would resemble one another and not lead to
any significant difference in model behaviour and the conclusions drawn. Support from this assumption comes from
a model run in which a globally uniform distribution of $TiO_2$ was assumed initially and which yielded very similar
results.
The first of these simulations included only the loss of $N_2O_5$ on $TiO_2$ particles with $\gamma(N_2O_5) = 0.005$, the upper limit
used in the modelling of Tang et al. (2014), which allows us to compare our results with their study. The second $TiO_2$
simulation also included the loss of $HO_2$ on $TiO_2$ particles with a $\gamma(HO_2) = 1$, as discussed above. Figure 8 shows
the impact of including heterogeneous loss of $N_2O_5$ on $TiO_2$ particles in the model.

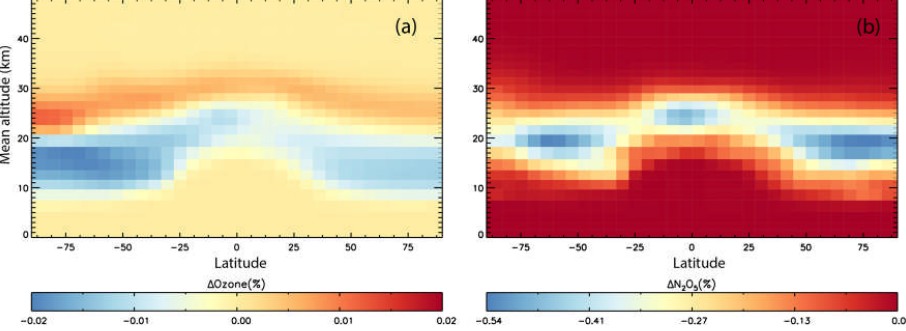


**Figure 8**. Annual mean change (%) in atmospheric $O_3$ (a) and $N_2O_5$ (b) calculated using the TOMCAT 3-D model
from inclusion of heterogeneous loss of $N_2O_5$ on $TiO_2$ particles for 2008 and using $\gamma(N_2O_5) = 0.005$.

$N_2O_5$ is decreased by up to 0.5% in the region of $TiO_2$ particles, which is assumed to follow the distribution of
sulphate particles after the Mt. Pinatubo eruption, in order follow the approach of previous modelling studies.
Assuming a globally uniform distribution initially yielded results that were very similar. Inclusion of uptake is only
a minor effect and considerably smaller than the impact of around -20% modelled by Tang et al. (2004) for the same
assumed $\gamma(N_2O_5) = 0.005$. The reasons for this are not clear, although it is noted that the effect modelled in our off-
line chemical transport model, with specified meteorology, is clearly confined to regions of high aerosol loading.





The impacts modelled in the nudged chemistry-climate model study of Tang et al. (2004) are not confined to the
region of high aerosol and even extend to the upper stratosphere. It is possible that their simulations, although nudged,
also include some dynamical feedback which enhances an otherwise small signal. Figure 8 also shows that the
resulting impact on $O_3$ is small with changes less than 0.02%. The model produces a region of slight decrease in the
very low stratosphere, with a region of slight increase above.
Figure 9 shows results from the simulation which also included the loss of $HO_2$ on stratospheric $TiO_2$ particles, and
using $\gamma(HO_2) = 1$ (a simulation using the measured value of $\gamma(HO_2) = 0.021$ at 295 K led to no impact on $O_3$ or $HO_2$).

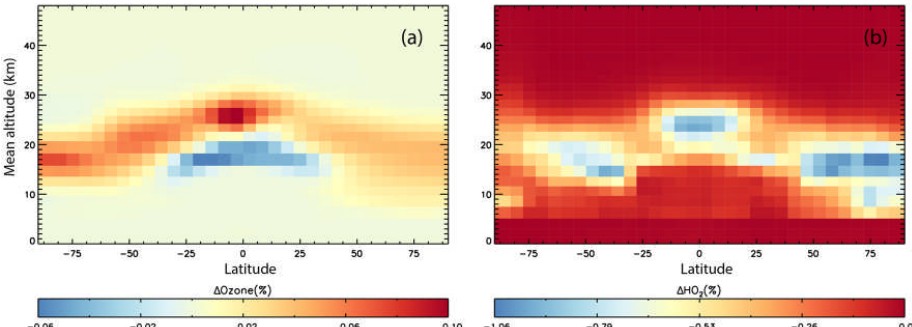


**Figure 9**. Annual mean change (%) in atmospheric $O_3$ (a) and $HO_2$ (b) calculated using the TOMCAT 3-D model
from inclusion of heterogeneous loss of $HO_2$ on $TiO_2$ particles for 2008, and using $\gamma(HO_2) = 1$.

It is evident that $HO_2$ loss due to heterogeneous reaction between $HO_2$ and $TiO_2$ particles in 2008 is < 1% and is
confined to the lower stratosphere where it is assumed $TiO_2$ particles are located.  Figure 9 also shows that the
subsequent effect of the $TiO_2$ particles on the $O_3$ concentrations through the effects of this reaction is also small
(< 0.1%), with a small decrease in the tropical upper troposphere/lower stratosphere and a small increase at all
latitudes in the lower stratosphere. This small effect of $TiO_2$ particles on stratospheric $HO_2$ and $O_3$ concentrations is
due to the reactive nature and short lifetime of $HO_2$. The species readily reacts with other gas phase species (e.g. $O_3$)
and so loss on $TiO_2$ surfaces does not compete significantly.

**4. Conclusions and further work**
The reactive uptake coefficients for the heterogeneous reaction of $HO_2$ onto $TiO_2$ particles were measured at different
RH and at room temperature for the first time using an aerosol flow tube reactor coupled with a sensitive FAGE $HO_2$
detection system. A range of $HO_2$ uptake coefficients on $TiO_2$ particles were measured varying from $\gamma(HO_2) = 0.021$
$\pm$ 0.001 to 0.036 $\pm$ 0.007 for RH over the range 11% to 66%, respectively.  The $HO_2$ uptake coefficient showed a





positive dependence on RH which correlated well with the number of monolayers of water adsorbed onto the $TiO_2$
particle. These results suggest a mechanism by which $HO_2$ adsorbs to the surface if the $TiO_2$ particle by forming
complexes with water molecules bound to bridging OH groups. As the number of water layers increases so does the
network of hydrogen bonds that stabilises $HO_2$ leading to a longer adsorption lifetime and increased $\gamma(HO_2)$. The
TOMCAT chemical transport model was used to evaluate the possible effects of $HO_2$ uptake (using an upper limit of
$\gamma(HO_2) = 1$) onto the surface of $TiO_2$ particles on the stratospheric concentrations of $HO_2$ and $O_3$. The amount of
$TiO_2$ used was chosen to achieve a similar cooling to that following the Mt. Pinatubo eruption, but the model predicted
a very small loss of both stratospheric $HO_2$ and $O_3$. $TiO_2$ possesses photocatalytic properties and water adsorbed
onto its surface may dissociate under stratospheric illumination providing a source of radicals (Chen et al.,
2012;Romanias et al., 2012). Production of OH and $HO_2$ from irradiated $TiO_2$ surfaces should be evaluated in future
studies to fully understand the consequences of injection of $TiO_2$ particles into the stratosphere.

**Acknowledgements**
We are grateful to the Natural Environment Research Council for funding a studentship (DRM) and for funding the
aerosol flow tube apparatus (grant number NE/F020651/1)). LKW, TI, PWS and and DEH are also grateful to the
NERC funded National Centre for Atmospheric Science for ongoing support. The TOMCAT modelling work was
supported by the EU StratoClim project (FP7 grant 603557). We thank Wuhu Feng (NCAS Leeds) for help with the
model. The model simulations were performed on the University of Leeds and N8 HPC system.



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
