# Peer review of "Heterogeneous reaction of HO2 with airborne TiO2 particles and its implication for"

_Atmospheric Chemistry and Physics, 2017_

## Referee Comment (RC1) · Anonymous Referee #1 · 27 Jun 2017

The work assesses the chemical reactivity of a potential candidate aerosol composition (titania) for geoengineering via the injection of stratospheric aerosol, a solar radiation management type approach to geoengineering. In particular the heterogeneous reaction of HO2 upon titania is assessed. The measured uptake coefficients show only negligible effect upon stratospheric ozone chemistry.

This is well executed science that combines rigorous laboratory work with global modelling. In particular, the kinetic data looks excellent (figures 2 and 3). A plausible explanation for the RH dependence is provided via a molecular model. The results are put in context and appropriate conclusions are reached. The manuscript is well writ-

ten. I have no major comments and I am happy for this to proceed to ACP post haste. I provide a few minor comments below.

Minor comments:

P2 L41 "showing sulphate aerosols to be unsuitable for solar radiation management" - this line is too strong. There are arguments why sulphate, or indeed any aerosol, should not be used. But if aerosol injection is to be considered, then sulphate is a significant contender by virtue of being nature's choice. i.e. in its favour is that there are natural experiments (volcanoes) with which to evaluate its performance and there are unlikely to be any surprises. The same cannot be said of titania.

P4 L96 - be more precise about the temperature that the experiment was conducted at, provide mean and standard deviation, rather than ∼293 K.

P4 L98 - what is the flow tube material? What wavelengths of light are omitted when the flowtube is shrouded?

P5 L117 - what form of titania was used (rutile, anatase etc.)? N.b. the surface characteristics of different forms of titania might have different surface reactivity.

P5 L136 - why do the particle diameters vary under different RH conditions? Titania is not significantly hygroscopic (as shown later in the paper with only monolayers of water being present at RH < 70%) so what is causing the size changes? Is it size dependent particle losses?

P11 L265 - define Vbridge

P14 L349 - only one SEM image is shown, presumably other images showed similar sphericity to the particle shown in Fig.7?

P366 L366 - note that a gamma of 1 implies every collision is reactive. This result maybe implies the laboratory work was not required?

P15 L388 - insert "to" in "...in order (to) follow..."

[Figure]

P16 L395 - the deviation between the consequences of N2O5 uptake between this study and the study of Tang et al. is interesting. A plausible explanation is provided but the authors are encouraged to discuss this discrepancy with the Tang et al. modellers (if they have not already).

---

## Short Comment (SC1) · 28 Jun 2017

I find this paper very interesting, and laboratory work on the topic of solar radiation management very promising. However, I have a comment regarding the discussion on sulfate geoengineering (SG) in the Introduction. In particular, while on line 34-35 effects on ozone depletion by SG are mentioned, in the following lines only old papers discussing ozone measurements after the Pinatubo eruption are mentioned, and from that comes the conclusion that those result are "showing sulphate aerosols to be unsuitable for solar radiation management". However, in recent years many papers regarding possible SG consequences on ozone depletion have been published. Even

if this would not change the scope of the paper itself, since it focuses on TiO2 particles, I would suggest adding some updated citations on the matter just to represent in an objective way the ongoing discussion of solar radiation management methods.

I include some possible papers to be cited regarding SG, its effectiveness and its indirect effects on ozone:

- Pitari, G., Aquila, V., Kravitz, B., Robock, A., Watanabe, S., Cionni, I., De Luca, N., Di Genova, G., Mancini, E., and Tilmes, S.: Stratospheric ozone response to sulfate geoengineering: Results from the Geoengineering Model Intercomparison Project (GeoMIP), J. Geophys. Res.-Atmos., 119, 2629–2653, 2014.

- Tilmes, S., Kinnison, D. E., Garcia, R. R., Salawitch, R., Canty, T., Lee-Taylor, J., Madronich, S., and Chance, K.: Impact of very short-lived halogens on stratospheric ozone abundance and UV radiation in a geo-engineered atmosphere, Atmos. Chem. Phys., 12, 10945–10955, doi:10.5194/acp-12-10945-2012, 2012

- Visioni, D., Pitari, G., and Aquila, V.: Sulfate geoengineering: a review of the factors controlling the needed injection of sulfur dioxide, Atmos. Chem. Phys., 17, 3879–3889, doi:10.5194/acp-17-3879-2017, http://www.atmos-chem-phys.net/17/3879/2017/, 2017.

---

## Referee Comment (RC2) · Anonymous Referee #2 · 15 Aug 2017

The authors present an aerosol flow tube and modeling study of HO2 uptake onto TiO2 particles. These particles have been proposed as solar geoengineering aerosols. The experiments were conducted very carefully but I question the authors' choice to study this reaction at room temperature when the relevant temperatures for solar geoengineering aerosol would be much lower. They did a careful study of the impact of adsorbed water on the reaction but this is irrelevant for stratospheric conditions. This dataset is valuable, even for our understanding of mineral dust chemistry in the troposphere, but the relevance for the stratosphere is in question.

In the comparison to existing datasets, instead of comparing the reactive uptake coefficient for HO2 on TiO2 to other species on TiO2, which is like comparing to apples to oranges, you can perhaps look at trends (ClONO2 on H2SO4 vs. TiO2 and HO2 on H2SO4 vs. TiO2, etc)

---

## Author Comment (AC1) · 26 Sep 2017

We thank the two anonymous referees and D. Visioni for their helpful comments.

We have replied to each point in turn below, the comments are given in italics first, with our response then given in normal type. Where appropriate, the text for the revised manuscript then follows in red type within speech marks.

**Anonymous Referee #1**

Clearly you are a Shakespearean scholar! Thanks for the adapted quotation from "A midsummer night's dream".

Comment 1: *P2 L41 "showing sulphate aerosols to be unsuitable for solar radiation management" - this line is too strong. There are arguments why sulphate, or indeed any aerosol, should not be used. But if aerosol injection is to be considered, then sulphate is a significant contender by virtue of being nature's choice. i.e. in its favour is that there are natural experiments (volcanoes) with which to evaluate its performance and there are unlikely to be any surprises. The same cannot be said of titania.*

Response 1: We entirely agree. We will soften the language. We discuss the eruption of Mt Pinatubo and the subsequent effects of the increased sulphate aerosol on observed stratospheric ozone concentrations. The revised text will be stated as follows:

"showing that sulphate aerosols could be unsuitable for solar radiation management"

Comment 2: *P4 L96 - be more precise about the temperature that the experiment was conducted at, provide mean and standard deviation, rather than ~293 K.*

Response 2:  All experiments were conducted at room temperature.  However, due to the time it takes to conduct an experiment not all experiments could be conducted on the same day. The average temperature across all experiments was 293 K. The standard deviation of the temperatures was 3 K. This will be stated in the new manuscript as follows:

"A schematic diagram of the experiment is shown in Figure 1, and all experiments were undertaken at room temperature (293 $\pm$ 3 K) and under normal laboratory levels of illumination."

Comment 3: *P4 L98 - what is the flow tube material? What wavelengths of light are omitted when the flow tube is shrouded?*

Response 3: The aerosol flow tube was made of borosilicate glass.  The experiments were conducted within a laser laboratory where there is no natural light present. Any light that could enter the flow tube would originate from the standard fluorescence strip lights in the laboratory. Therefore the wavelengths of light omitted wen the flow tube is shrouded would be within the visible wavelength region.

"In this investigation a borosilicate glass aerosol flow tube coupled with a sensitive $HO_2$ detector based on chemical conversion followed by laser-induced fluorescence detection of

OH (George et al., 2013) has been used to study the kinetics of the heterogeneous reaction of HO2 with airborne $TiO_2$ nanoparticles at different RH."

Comment 4: *P5 L117 - what form of titania was used (rutile, anatase etc.)? N.b. the surface characteristics of different forms of titania might have different surface reactivity.*

Response 4: We specify that the titania used is a blend known as Degussa (P5 L117), which consists of 80% anastase and 20% rutile. This particular blend is often used within laboratory experiments and is readily available.

"A solution of $TiO_2$ (Aldrich Chemistry 718467, 99.5% Degussa i.e. a blend of $TiO_2$ polymorphs 80% anatase and 20% rutile, 5 g in 500 ml of Milli-Q water) was placed in a commercial atomizer (TSI 3076) in order to produce a 1 L min$^{-1}$ flow entrained with $TiO_2$ particles, referred to as the aerosol flow."

Comment 5: *P5 L136 - why do the particle diameters vary under different RH conditions? Titania is not significantly hygroscopic (as shown later in the paper with only monolayers of water being present at RH < 70%) so what is causing the size changes? Is it size dependent particle losses?*

Response 5: We are unsure ourselves to the precise reason for this, however, we agree that the cause may be associated with differences in aerosol transmission through the experimental setup at different RH before aerosols reach the flow tube. However, by placing the SMPS before and after the aerosol flow tube we have shown that there is not significant loss of aerosol in the flow tube. Another possible reason could be that aerosol generation could change slightly from day to day due, for example, to a blockage in some parts of the delivery set up (the measurements of the size distribution at the two RH were not done in the same day).

Comment 6: *P11 L265 - define Vbridge*

Response 6: $V_{bridge}$ is defined as an oxygen vacancy within the main body of the text on P12 L272. We will define it at first usage here in the caption.

"Figure 6. Simplified diagram of the important steps of HO2 reactive uptake onto the surface of $TiO_2$. Grey circles = Ti, blue circles = O, orange circles = H, green circles = oxygen vacancy ($V_{bridge}$), solid black lines = chemical bond and dashed lines = hydrogen bond."

Comment 7: *P14 L349 - only one SEM image is shown, presumably other images showed similar sphericity to the particle shown in Fig.7?*

Response 7: Many other images were taken which do indeed show that all particles are spherical, however, due to the method of sample preparation for the SEM it was difficult to find a particle in isolation as most were agglomerated together.

Comment 8: *P366 L366 - note that a gamma of 1 implies every collision is reactive. This result maybe implies the laboratory work was not required?*

If the measured $HO_2$ uptake coefficient at room temperature is used, then the model shows that this reaction has negligible effect on stratospheric $O_3$ or $HO_2$. However, as outlined in the paper, the uptake coefficient is likely to be much higher at the very low temperatures found in the stratosphere. It is not possible to measure the uptake coefficient at these temperatures, and so a value of 1 (the highest possible) was assumed. The impact on $O_3$ and $HO_2$ is still small, but there is an effect. However, although the experimental values are not used in the model, the experiments are still very valuable, as they enable the mechanism for uptake of $HO_2$ on $TiO_2$ to be explored. In addition, $TiO_2$ provides a good model for mineral dusts and hence these experiments aid to help the interpretation of uptake of trace gases onto mineral dusts. The use of $TiO_2$ as an environmental catalyst is also not confined to the stratosphere and the dataset may be relevant for tropospheric applications of $TiO_2$ aerosols, for example in its use for the surface oxidation of pollutants such as VOCs and $NO_2$.

Comment 9: *P15 L388 - insert "to" in "...in order (to) follow..."*

Response 9: This will be amended.

"$N_2O_5$ is decreased by up to 0.5% in the region of $TiO_2$ particles, which is assumed to follow the distribution of sulphate particles after the Mt. Pinatubo eruption, in order to follow the approach of previous modelling studies."

Comment 10: *P16 L395 - the deviation between the consequences of N2O5 uptake between this study and the study of Tang et al. is interesting. A plausible explanation is provided but the authors are encouraged to discuss this discrepancy with the Tang et al. modellers (if they have not already).*

Response 10: Prior to submission we had a brief verbal discussion with Dr James Keeble (University of Cambridge) who performed the modelling work in Tang et al (2014). Based on that discussion we suggested possible deviations in the modelling approaches which may be responsible for the differences in the $N_2O_5$ response. Following this review comment we have been in further contact with Dr Keeble. This has not led to any definitive conclusions, as that would require significant extra work. However, along the lines suggested by the reviewer we have agreed to keep in contact and investigate these differences in future work. We will mention the need for further work in the text.

**Anonymous Referee #2**

Comment 1: *The experiments were conducted very carefully but I question the authors' choice to study this reaction at room temperature when the relevant temperatures for solar geoengineering aerosol would be much lower. They did a careful study of the impact of adsorbed water on the reaction but this is irrelevant for stratospheric conditions. This dataset is valuable, even for our understanding of mineral dust chemistry in the troposphere, but the relevance for the stratosphere is in question.*

Response 1: We acknowledge in the manuscript that the experiments were not conducted at stratospherically relevant temperatures, and state the experimental difficulties which prevent us from performing experiments at such temperatures (P3 L86, P14 L365). However, this represents the first measurements of $HO_2$ onto $TiO_2$ aerosols, and stratospheric $TiO_2$ injection has been suggested as a mitigation strategy. Both previous studies of $N_2O_5$ and $ClONO_2$ onto $TiO_2$ aerosols by Tang et al. were also conducted only at room temperature for the same reason. Although the application is framed within the context of stratospheric injection of aerosols for solar radiation management schemes, we agree that the dataset is also valuable for our understanding of mineral dust chemistry in the troposphere and evaluating the feasibility of other environmental catalytic applications of $TiO_2$ within the troposphere such as removal of VOCs and $NO_2$.

Comment 2: *In the comparison to existing datasets, instead of comparing the reactive uptake coefficient for HO2 on TiO2 to other species on TiO2, which is like comparing to apples to oranges, you can perhaps look at trends (ClONO2 on H2SO4 vs. TiO2 and HO2 on H2SO4 vs. TiO2, etc).*

Response 2: We accept this point although it is still useful to compare the behaviour of $HO_2$ towards $TiO_2$ aerosols with the very few other studies of uptake of trace gases onto $TiO_2$ aerosols, as it gives the opportunity to compare and contrast the likely mechanisms for uptake of trace gases onto $TiO_2$ aerosols. There have been very limited studies of $HO_2$ uptake onto solid surfaces which have been coated with $H_2SO_4$ aerosol material (Gershenzon et al., Faraday Discussions, 1995, 100, 83-100) and we will include a comparison of these experiments with uptake onto $TiO_2$. The modified text is as follows:

"Measurements of $\gamma(HO_2)$ have been made onto sulphuric acid aerosols and thin films. These values are not consistent and range from > 0.1 to < 0.01 (Cooper and Abbatt, 1996; Thornton and Abbatt, 2005; Hanson et al., 1992; Gershenzon et al., 1995). The most recent measurement of $\gamma(HO_2)$ on aqueous phase sulphuric acid aerosols was conducted at 35% RH (Thornton and Abbatt, 2005). That study estimated a value of $\gamma(HO_2) = 0.006 \pm 0.004$, lower than onto $TiO_2$ aerosols. Whilst the aerosols in that study are unlike sulphuric acid aerosols in the stratosphere, formed via condensation of sulphuric acid vapour onto existing solid aerosols, the relatively low value of $\gamma(HO_2)$ measured is consistent with the likely low partitioning of $HO_2$ to its more reactive conjugate base, $O_2^-$, and a lower solubility of $HO_2$ in aerosols with a low pH. The study concludes that heterogeneous uptake of $HO_2$ onto sulphuric acid aerosols would show a strong negative temperature dependence driven by the temperature dependence of the Henry's Law coefficient. It is likely that at temperatures typical of the lower stratosphere (205 – 215 K) $\gamma(HO2)$ onto sulphuric acid particles approach 1 (Gershenzon et al., 1995)."

References:

Cooper, P. L., and Abbatt, J. P. D.: Heterogeneous interactions of OH and $HO_2$ radicals with surfaces characteristic of atmospheric particulate matter, Journal of Physical Chemistry, 100, 2249-2254, 1996.

Thornton, J., and Abbatt, J. P. D.: Measurements of $HO_2$ uptake to aqueous aerosol: Mass accommodation coefficients and net reactive loss, J. Geophys. Res.-Atmos., 110, 10.1029/2004jd005402, 2005.

Hanson, D. R., Burkholder, J. B., Howard, C. J., and Ravishankara, A. R.: Measurement of hydroxyl and hydroperoxy radical uptake coefficients on water and sulfuric acid surfaces, The Journal of Physical Chemistry, 96, 4979-4985, 10.1021/j100191a046, 1992.

Gershenzon, Y. M., Grigorieva, V. M., Ivanov, A. V., and Remorov, R. G.: $O_3$ and OH sensitivity to heterogeneous sinks of $HO_x$ and $CH_3O_2$ on aerosol particles, Faraday Discussions, 100, 83-100, 1995.

**Comment from Daniele Visioni**

Comment 1: *I find this paper very interesting, and laboratory work on the topic of solar radiation management very promising. However, I have a comment regarding the discussion on sulfate geoengineering (SG) in the Introduction. In particular, while on line 34- 35 effects on ozone depletion by SG are mentioned, in the following lines only old papers discussing ozone measurements after the Pinatubo eruption are mentioned, and from that comes the conclusion that those result are "showing sulphate aerosols to be unsuitable for solar radiation management". However, in recent years many papers regarding possible SG consequences on ozone depletion have been published. Discussion paper if this would not change the scope of the paper itself, since it focuses on TiO2 particles, I would suggest adding some updated citations on the matter just to represent in an objective way the ongoing discussion of solar radiation management methods. I include some possible papers to be cited regarding SG*

Response 1: This comment is helpful, thank you, and the introduction will be amended accordingly to include citation of the 2017 Review on sulfate geoengineering. The following text will be added:

"The impact of sulphate geoengineering to stratospheric ozone concentrations is projected to decrease in time as concentrations of Br and Cl containing atmospheric species are expected to fall, so much so that beyond 2050 the additional available surface area provided by sulphate geoengineering is predicted to enhance conversion of NOx (NOx = NO + $NO_2$) to $HNO_3$ resulting in an increase of stratospheric ozone (Visioni et al., 2017)."

Reference:
Visioni, D., Pitari, G., and Aquila, V.: Sulfate geoengineering: a review of the factors controlling the needed injection of sulfur dioxide, Atmos. Chem. Phys., 17, 3879-3889, 10.5194/acp-17-3879-2017, 2017.